# Prelocalization and Leak detection in drinking water distribution network using modeling-based algorithms: Case study: The city of Casablanca (Morocco)

Faycal Taghlabi [1], Laila Sour [1], Ali Agoumi [2]

[1] Laboratory of Processes and Environment, Faculty of Science and Technology of Mohammedia, Hassan II university of Casablanca, 28806, Morocco

[2] Laboratory of Civil, hydraulic Engineering, Environment and Climate, Hassania School of Public Works, Casablanca,20200, Morocco

*Correspondence to*: Faycal Taghlabi (ftaghlabi@yahoo.fr)

**Abstract.** The role of a drinking water distribution network ( DWDN) is to supply high-quality water at the necessary pressure at various times of the day for several consumption scenarios. Locating and identifying water leakage areas has become a major concern for managers of the water supply, to optimize and improve constancy of supply. In this paper, we present the results in the field research conducted to detect and locate leaks in the ( DWDN) focusing on the resolution of the Fixed And Variable Area Discharge (FAVAD) equation by use of the prediction algorithms in conjunction with hydraulic modeling and the Geographical Information System (GIS).The leak localization method is applied in the oldest part of Casablanca. We have used, in this research, two methodologies in different leak episodes: (i)The first episode is based on a simulation of artificial leaks on the MATLAB platform using the EPANET code to establish a database of pressures that describes the network's behaviour in the presence of leaks. The data thus established has fed into a machine learning algorithm called Random Forest, which will forecast the leakage rate and its location in the network; (ii)The second was field-testing a real simulation of artificial leaks by opening and closing of hydrants, on different locations with a leak size of 6l/s and 17 l/s. The two methods converged to comparable results, the leaks position is spotted within a 100 m radius of the actual leaks.

Keywords: FAVAD; WDN; Leak localization; prediction; Epanet.

## 1 Introduction

Climate Change is a major global issue, more and more important on the international scene. It affects all components of the hydrological cycle. The situation of water resources in Morocco is already critical with a state of water scarcity forecasted for 2020. This problem is accentuated by the effects of  Climate Change and may hinder any further sustainable development. The expected Climate Change for Morocco would have direct and indirect harmful consequences on the water resources potential, in terms of both quantity and quality, on the water demand and on the efficiency of use of this resource by the different users. An anticipation of the adaptation to the effects of this Climate Change must pass by the valorization of the use of the resources and especially the minimization of the water losses. In this regard, in Moroccan urban areas, drinking water distribution

networks have particularly low yields. The location and prioritization of leaking areas is a major concern for the public authorities to optimize the use of water resources, reduce losses and improve continuity of service.

To guarantee the high-level service of pressure, the detection and repair time of leaks is certainly the most common factor used in the analysis of decreases in contract pressures.

For most of the time, before starting a leak detection campaign in a Discrete Hydraulic Sector (DHS) we start with the analysis of the flows into and out of the sector, in particular the minimum night flow (MNF) between 2:00 AM and 4:00 AM, as well as the volumes of major consumers (Alkasseh et al., 2013).

In the literature it is possible to detect leaks in the DHS, usually the leakage rate is permanent over time, if the DHS records an increase in night flow, this increase should also appear during normal consumption time (Oasen, 2015a). According to

research by Farley et al., (2008) An increase in minimum night flow can be used for targeting all "DHS"where leakage is more likely.  It is therefore possible to detect leaks in a DHS by making a hydraulic balance between the volume of billed consumption and the volume distributed, by comparing the expected demand and the actual water consumption (Bakker, 2014). Once new leaks by DHS are identified, various techniques are used to locate the leaks. Acoustic leak-detection is a technique which has evolved a lot in recent years and is developing rapidly (Farley, 2003). Some of these techniques require partitioning

a DWDN into smaller DHS, by closing certain valves on the network, which can sometimes shutdown the system (Colombo, 2009).

In addition, various research projects noted that it is difficult to apply the leak-detection to certain areas due to the complexity of isolating and partitioning (Andrea et al., 2011). Through the applied works in modeling leakage, in particular, those of Babel et al., (2009) and Sebbagh et al., (2017), a reduction in pressure at the inlet of DHS, induces a reduction in leakage rate. For

Al-Ghamdi et al (2011), a 25% reduction in pressure contributes to a leakage flow reduction of about 25% for a 50% rigid 50% plastic network.

Our approach, as we will see through the following paragraphs, is to do a virtual leak search without partitioning a DWDN into smaller DHS.

The deficiency of leak management is one of the key problems, given its impacts on production cost and resource exhaustion.

The scope of this paper will be mainly focused on the application of the two approaches in two different leak events. The case study is a pilot sector in the city of Casablanca (Morocco), which covers around 24 000 inhabitants as displayed in Fig. 1

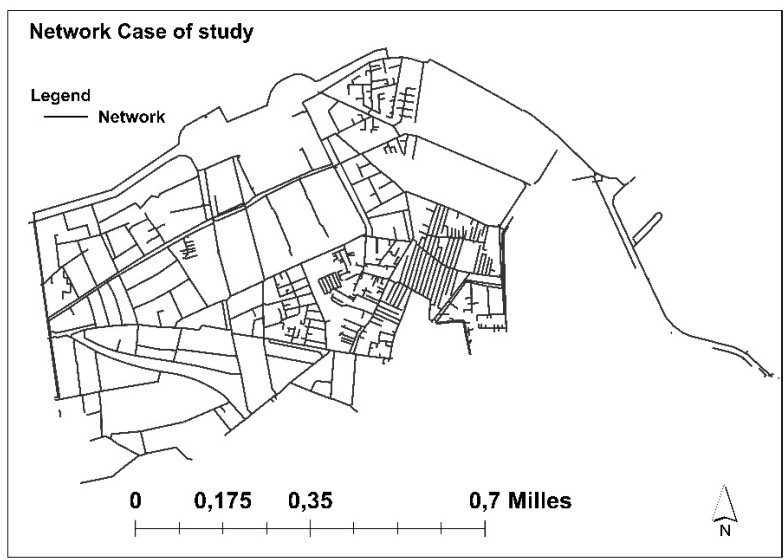

**Figure 1 Delimitation of the study area.**

The study area is a micro modulated sector with a single critical pressure point that is continuously monitored. It has three inlets, 493 nodes and 42 km of pipes. Each of the three inlet of the zone has its own flow meter in diameter 300, and the hydraulic model was calibrated at each inlet.

The network flow and pressure are monitored through flowmeter in diameter 300 mm and pressure sensors at each inlet.

The following figure 2 shows the daily average of measured water demand

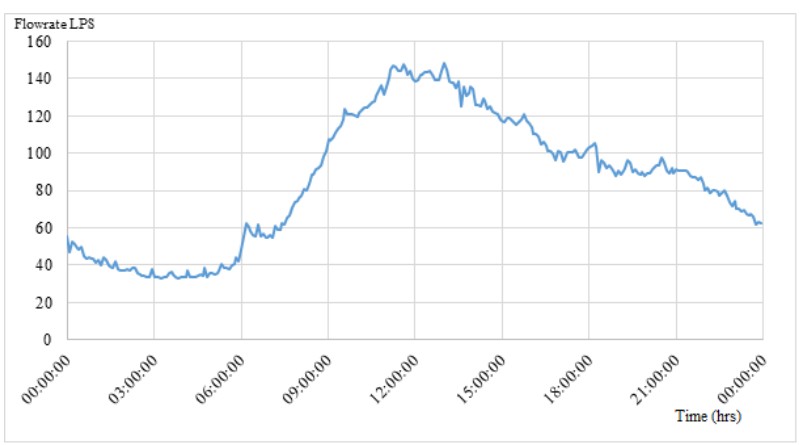


**Figure 2 Daily average water demand of the study area.**

The figure above shows a significant minimum night flow around 40 LPS between 2:00 and 4:00 AM, with a linear loss index (LLI) of 54 m3/day/km which implies a high probability of the presence of physical losses.

Figure 3 reveals that the network consists mainly of greycast iron (40%). The average age is 40 years, increasing vulnerability and promoting leaks.

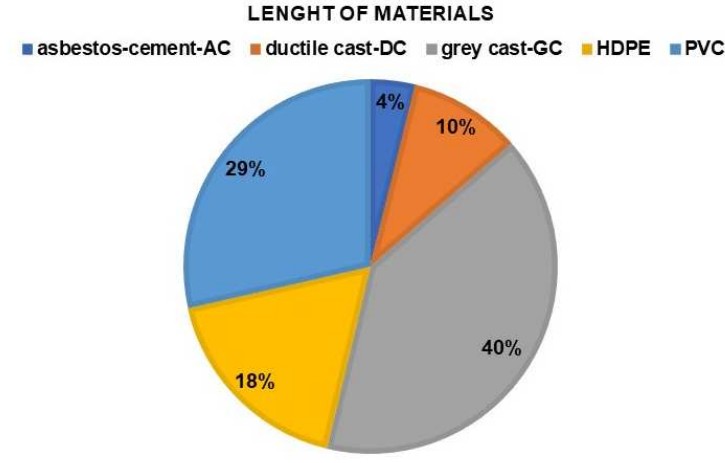

**Figure 3 Types of materials constituting the network**

## 2 Materials and methods

**2.1 Software**

EPANET is a free software developed by the US Environmental Protection Agency (U.S. EPA). From the representation of the distribution network (nodes, pipes, tank, valves, pump, etc.), it allows the hydraulic balancing of the network by the calculations of pressure losses, flow velocity, flow in the pipes and pressure at the nods. (Rossman, 2000).

In practice EPANET is used by water utilities (EPA 2005) and in literature (Farina et al., 2014).

The basic demand for the hydraulic modeling software EPANET 2.0 is defined as a water output at each node, We consider that there are two main methods to simulate a water leak in EPANET, as an additional demand; or even water flow rate through a Valve, the formula to calculate the head loss (Darcy-Weisbach) was used with the default values for the roughness (Brown, 2002).

**2.2 Method**

(i) Relationships between pressure and leakage rates in distribution networks.

Pressure management not only involves reducing pressure, but also other pressure control and optimization methods without compromising customer service. A definition of pressure management in its broadest sense is given by Thornton et al., (2005), "pressure management is about controlling the pressure of the system to achieve a level of optimal service, to ensure an efficient supply to consumers while avoiding the unnecessary excesses of this pressure which would unduly increase leaks".

Water utilities often take to design their distribution networks the minimum pressure that occurs at the critical point at maximum demand. Understanding this concept is of great importance as pressure regulation can significantly reduce leakage without compromising the level of customer service.

Empirical research has repeatedly shown that the Fixed and Variable Area Discharges (FAVAD) principle, which demonstrates the fact that most discharges from pressurized pipelines vary with pressure to a greater or lesser extent. This concept, via the

definition of an exponent N1, defines the relationship between the leakage rate and the pressure in case of pressure modulation According to (Lambert, 2000) and (Rozental, 2010) the relationship between pressure and flow leaks is given by Eq. (1):

$$L1 = L0 \times (P1/P0)^{N1} \tag{1}$$

where L1 and P1 are respectively the leakage rate and the average pressure in the DHS during the day, L0 and P0 are respectively the leakage rate and average pressure at the minimum night flow (MNF) time, between 2:00 AM and 4:00 AM.

According to Al Ghamdi, (2011) et Cobacho et al., (2014), the main method for representing leaks in a hydraulic network model is through adding a leakage valve for each node, the emitter parameter is used to model flow rate through a valve. These emitters devices permit the modelling of flow evacuated to the atmosphere through a nozzle.

the equation below represents the concept of FAVAD, through a flow rate, pressure and emitter coefficient Eq. (2):

$$Qleak\ j = C \times Pj^{N} \tag{2}$$

where Qleak is the flow rate at node j, C is the emitter coefficient, P pressure at node j and N is the pressure exponent.

The exponent N of the above equation varies according to the material of the pipeline (mainly its elasticity), for a circular opening on a rigid pipe (cast iron, steel), N1 is of the order of 0,5 whereas it reaches 1,5 or more for longitudinal slots on plastic materials (PVC, PEHD). However, international feedback shows a variation of N1 between 0.36 and 2.95 depending on the networks experienced, as shown in Fig. 4(Rozental, 2010). Figure 4 illustrates the influence of N1 the exponent emitter

on the impact of pressure reduction on leakage rate.

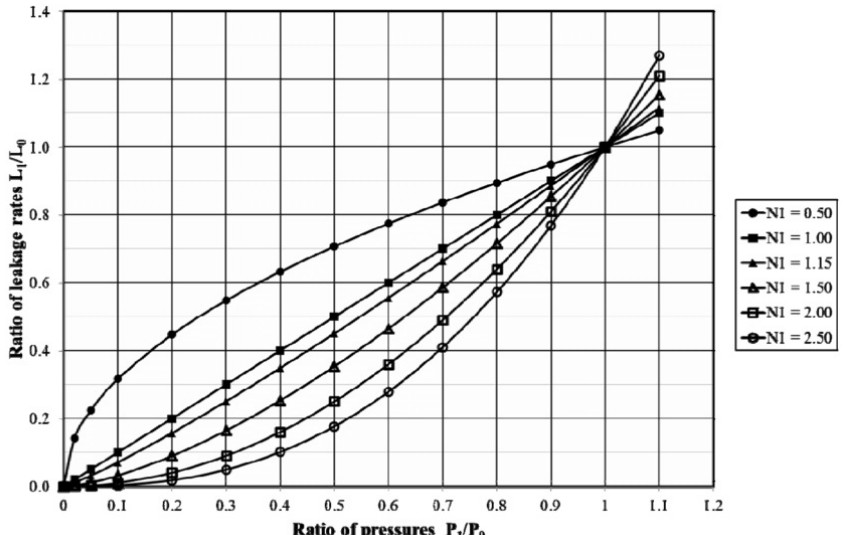

**Figure 4 Relationships between pressure and leakage rate using the N1 Approach (Rozental, 2010)**

The data collection relative to the various components of the drinking water networks (pipes, reservoirs, well, drilling, pumps, valves) is made by means of shapefile exported from the database cart@jour. It's an interface GIS (Geographical Information System) available for consultation in intranet which includes three networks managed by the drinking water operators as well as all of the hydraulic structure which constitutes them. The interface allows extracting all the desired layers while geometrically targeting the study area (zone of study).The following Table illustrates the roughness values used during modeling (Chadwick et al., 2013).

**Table 1 Roughness coefficient of materials (values from Chadwick et al., 2013)**

| Material | Age [year] | Darcy-Weisbach roughness coefficient [mm] |
|---|---|---|
| AC | 0-10 | 0.10 |
| | 10-40 | 0.15 |
| | >40 | 1.2 |
| All plastics: HPE, PVC, ZPE | 0-10 | 0.05 |
| | 10-40 | 0.10 |
| | >40 | 0.15 |
| ST | 0-10 | 0.3 |
| | 10-40 | 1.0 |
| | >40 | 2.0 |

The elevations are extracted from the Digital Elevation Model (DEM) layer and automatically assigned to network nodes.

The annual average consumption for 2017 in addition, are distributed into each node in the model according to the geographical distribution of subscribers within the tour. Once the network template is prepared using ArcGIS, it is transferred to the EPANET software in .inp file.

The EPANET hydraulic simulation model calculates node pressure and pipe flow for a fixed reservoir level and variable water demands over time and space. It predicts the dynamic hydraulic behaviour within a drinking water distribution system operating over an extended period of time. The pressure changes due to discharge, pressure calculating and changes according to base demand and daily consumption patterns at each node. The pressure drop during a peak period of consumption is due to higher consumption patterns in the DHS (The pattern provides multipliers that are applied to the Base Demand to determine actual demand in a given time period).

The Figure 5 bellow shows an example of daily consumption patterns in the study area.

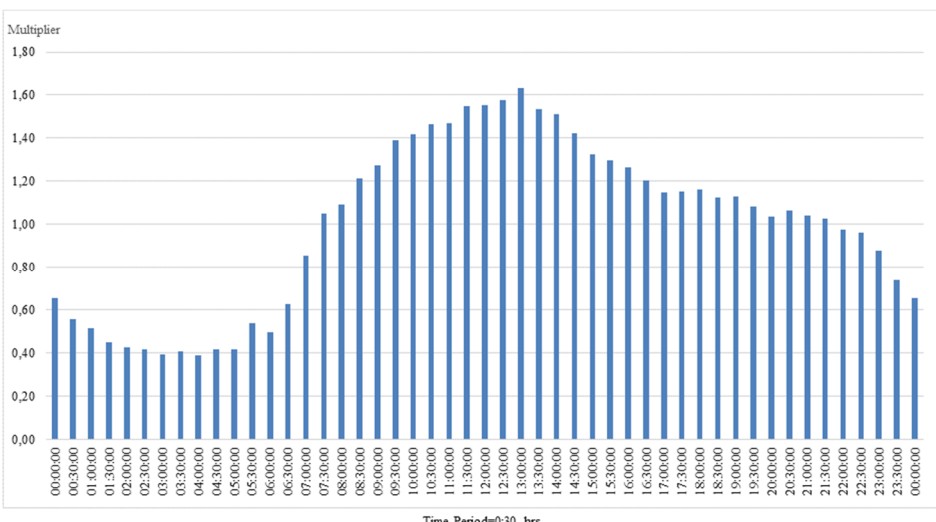

**Figure 5 daily consumption patterns in the study area**

Several simulations using the Epanet software were used to determine the roughness coefficients of the pipes to obtain calculated pressures which indicate the actual pressures in different nodes of the hydraulic system.

Different factors are attributable to the uncertainty of the pressure measured on the network, the most important of which is the fluctuation of the pressure in the network. According to the pressure differences observed on the ground when determining roughness, according to the work of Paquin et al., (2000) this error is estimated at 0.3 mCE. Regarding the error in modeling

the reference pressures, a value of 0.2 mCE is considered realistic. The total uncertainty related to the pressure difference between the measured pressure and the reference pressure is therefore 0.5 mCE.

Various factors including the measuring devices and the state of equilibrium of the network can influence the accuracy of the value of the roughnesses measured in the field, as well as for the estimation of roughness for the other conduits of the network. The following figure 6 presents 3 curves which correspond to the pressure drop caused by a leak in the middle of a pipe without

a junction, one with roughness as measured in the field, and two others with limit values taking into account a +/- 10% error related to this parameter ($\mathcal{E} + 10\%$, $\mathcal{E} -10\%$).

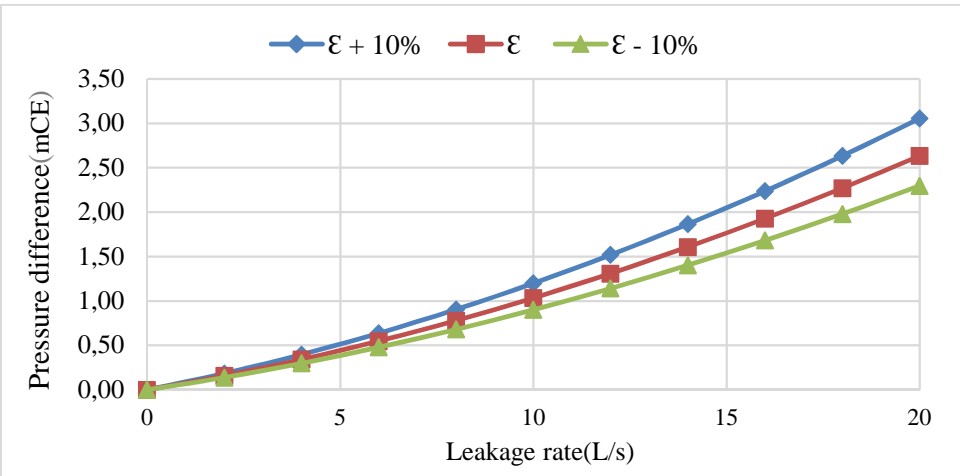

**Figure 6 Influence of the error on the roughness of the pipe**

When the roughness is increased, the pressure difference caused by a leak is greater. Conversely, when the pipes are smoother (plastic), the pressure difference decreases. Leak detection by the method studied therefore seems more promising on networks whose hydraulic capacity is weakened by corrosion rates.To solve this localization of leakages we will use the random forest algorithm (R Learning and Prediction Algorithm).

Algorithm 1 (Section 2,3,2) illustrates briefly the suggested leakage detection procedure.

The leakage localization methodology displayed in Fig. 8is based on data mining algorithms, the starting point of the algorithm is the learning of the data obtained by simulation using EPANET simulator. Then four training data elements are used to predict the location of the leak:

- The distance between simulation node and the sensor
- The leakage flow
- Emitter Coefficient
- Pressure at the sensors

In order to estimate the network flow, the leakage outflow at each node, and the distance of leak from sensors, a leakage model is implemented within a classical hydraulic simulation model (EPANET).

There are two ways to model a water leak in a hydraulic network model EPANET (i) as an additional nodal demand, and (ii) by adding a leak valve to each as shown in figure 7 below.

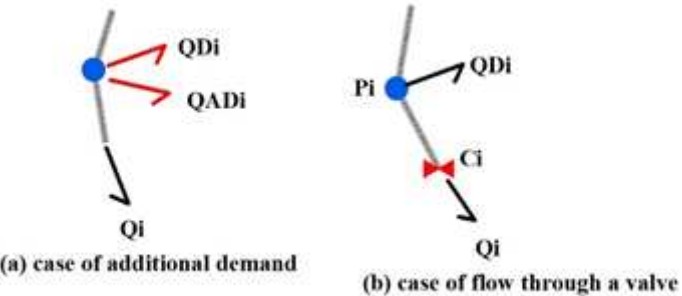

(a) case of additional demand          (b) case of flow through a valve


**Figure 7 Ways to model a water leak in a hydraulic network model**

where QDi is the node base demand (consumption), QADi is the node additional demand, Pi is the node pressure, Ci is the leak valve coefficient and Qi is the leakage flow.

According to Cobacho et al., 2014 the best way to represent leakage in a hydraulic network model is not by means of an

additional demand, but rather by adding a leak valve to each node, the dynamic behavior of leakage is appreciated in this case rather than the case as an additional demand.

In Epanet, the closest element to a leak valve is the emitter, which presents an open valve to the atmosphere, the emitter behavior equation is as shown above in Eq 2.

The emitter coefficient is placed at the junctions, 4 emitter coefficients ( 0.8, 1.6, 2.4 and 3.2)  were used in this simulation

step at each node, which correspond respectively to approximate flow leaks: 5, 10, 15 and 20 l/s, the objective of this simulation step is to vary the leak rate at nodes, to calculate and generate the new profile of pressures at each changes in flow rate.

The simulation results obtained constitute the training data for our machine learning model.

(ii) Leak simulation in Epanet:

This training data is obtained by using EPANET toolkit available in [https://github.com/OpenWaterAnalytics/EPANET-

Matlab-Toolkit] for use in MATLAB. It is an open-source software that provides programming interface for EPANET within MATLAB framework. It is easy to modify, simulate and plot the result produced by EPANET libraries (Eliades et al., 2016). Figure 9 reveals that 493 nodes were defined to simulate leaks with different emitter coefficients: 0.8, 1.6, 2.4 and 3.2, which corresponds approximately to leak rates of 5, 10, 15 and 20 l/s as per Eq. (2)

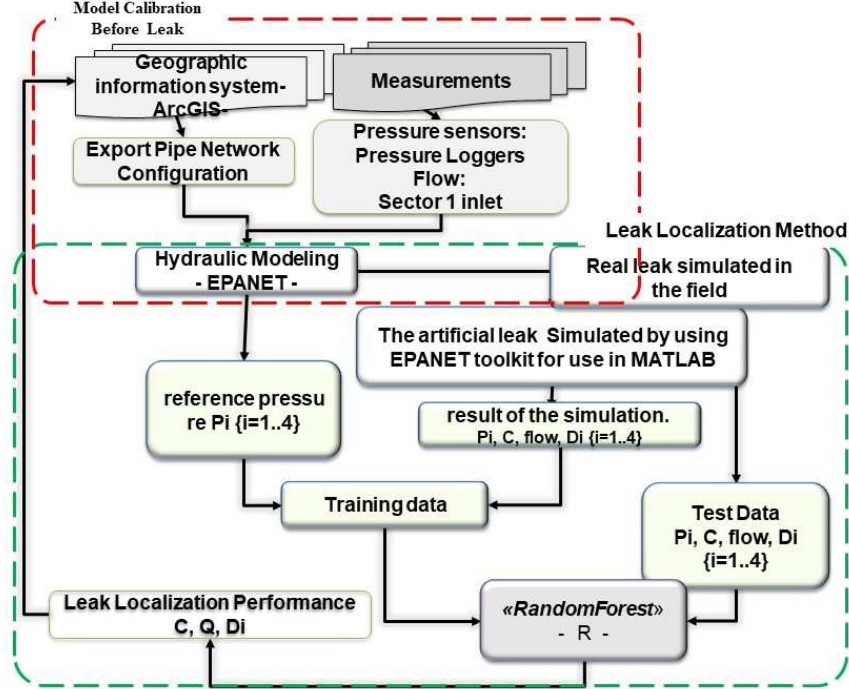

**Figure 8 Leak pre-localization procedure**

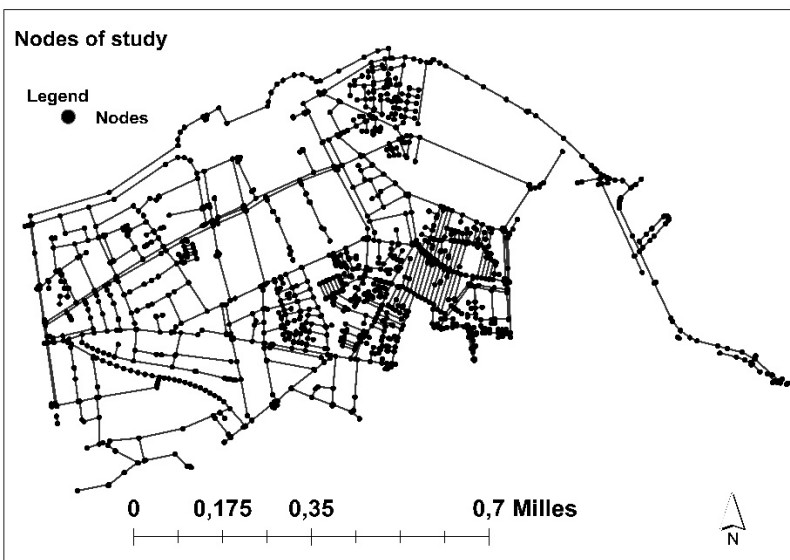

**Figure 9 Leak simulation nodes in sector 1**

Thus, at each node, we simulated 4 leaks with emitter coefficients equal to 0.8, 1.6, 2.4 and 3.2, for a total of 1972 simulations. The purpose of this 1972 simulations was to stimulate the pressure behavior at each node, the area contains 493 nodes, thus,

we simulated 4 leaks with emitter coefficients equal to 0.8, 1.6, 2.4 and 3.2. This makes a total of 1972 simulations. Each time, the leakage rate, the emitting coefficient, the location of the leak, the distances from the four sensors P14, P57, P58 and PC are noted, as well as the maximum, minimum and average pressures at these measuring points. the authors would like to record the maximum data of pressure variations in the case even if the leaks are triggered in the looped far part on the network. A completely new base for data learning. The results are reported in Table 2.

**Table 2 Result of Model Leak Simulations**

| Data | Node 1 | Node 2 | Node 3 | Node 4 | Node 5 | Node 6 | Node 7 | Node 8 | …… |
|------|--------|--------|--------|--------|--------|--------|--------|--------|------|
| Pmax1 | 38,91 | 38,85 | 38,78 | 38,70 | 38,91 | 38,85 | 38,78 | 38,70 | …… |
| Pmin1 | 37,33 | 37,16 | 36,98 | 36,80 | 37,33 | 37,16 | 36,98 | 36,80 | |
| Pmoy1 | 38,39 | 38,29 | 38,17 | 38,04 | 38,39 | 38,29 | 38,17 | 38,04 | |
| Pmax2 | 37,48 | 37,43 | 37,36 | 37,29 | 37,48 | 37,43 | 37,36 | 37,29 | |
| Pmin2 | 35,66 | 35,50 | 35,34 | 35,16 | 35,66 | 35,50 | 35,34 | 35,16 | |
| Pmoy2 | 36,87 | 36,77 | 36,65 | 36,53 | 36,87 | 36,77 | 36,65 | 36,53 | |
| Pmax3 | 36,89 | 36,84 | 36,78 | 36,71 | 36,89 | 36,84 | 36,78 | 36,71 | |
| Pmin3 | 34,91 | 34,75 | 34,59 | 34,42 | 34,91 | 34,75 | 34,59 | 34,41 | |
| Pmoy3 | 36,20 | 36,11 | 35,99 | 35,87 | 36,20 | 36,11 | 35,99 | 35,87 | |
| Pmax4 | 34,51 | 34,46 | 34,40 | 34,33 | 34,51 | 34,46 | 34,40 | 34,33 | |
| Pmin4 | 32,17 | 32,02 | 31,86 | 31,69 | 32,17 | 32,02 | 31,86 | 31,69 | |
| Pmoy4 | 33,66 | 33,56 | 33,45 | 33,33 | 33,66 | 33,56 | 33,45 | 33,33 | |
| C | 0,8 | 1,6 | 2,4 | 3,2 | 0,8 | 1,6 | 2,4 | 3,2 | |
| Q | 4,96 | 9,90 | 14,83 | 19,73 | 4,96 | 9,90 | 14,83 | 19,73 | |
| D1 | 478,21 | 478,21 | 478,21 | 478,21 | 475,51 | 475,51 | 475,51 | 475,51 | |
| D2 | 399,55 | 399,55 | 399,55 | 399,55 | 483,06 | 483,06 | 483,06 | 483,06 | |
| D3 | 486,43 | 486,43 | 486,43 | 486,43 | 483,06 | 483,06 | 483,06 | 483,06 | |
| D4 | 729,73 | 729,73 | 729,73 | 729,73 | 726,19 | 726,19 | 726,19 | 726,19 | |

Or :

•        D1: the distance from the simulation node to the P14

•        D2: the distance from the simulation node to the P57

•        D3: the distance from the simulation node to the P58

•        D4: the distance from the simulation node to the PC

•        Q: the leakage

- C: emitter coefficients
- 205 • Pmax 1: The maximum pressure during low consumption period at P14
- Pmin 1: the minimum pressure in rush hour period at P14
- Pmoy 1: average daily pressure at P14
- Similarly, for the indices 2,3 and 4 which correspond respectively to the P57, P58 and PC,

The table thus constructed constitutes the input data of our algorithm

## 2.3 Location and simulation of leaks on the ground

### 2.3.1 leak pre-locating

In this section we remind the existing techniques of leak detection. The technique used to search preventive leakage on distribution networks is organized around three distinct but complementary operations: sectoring, pre-location, followed by localization. These methods must be adapted according to the dimensions and the degree of knowledge of the targeted. The following figure 10 summarizes the existing pre-localization stages, these stages make it possible to go from hundreds of km of network to tens of m, by proceeding by elimination: In the identified leaky zones, the night flow of sectors is measured then sub-sectors to identify leaking sections and precisely guide acoustic detection and then the location of leaks.

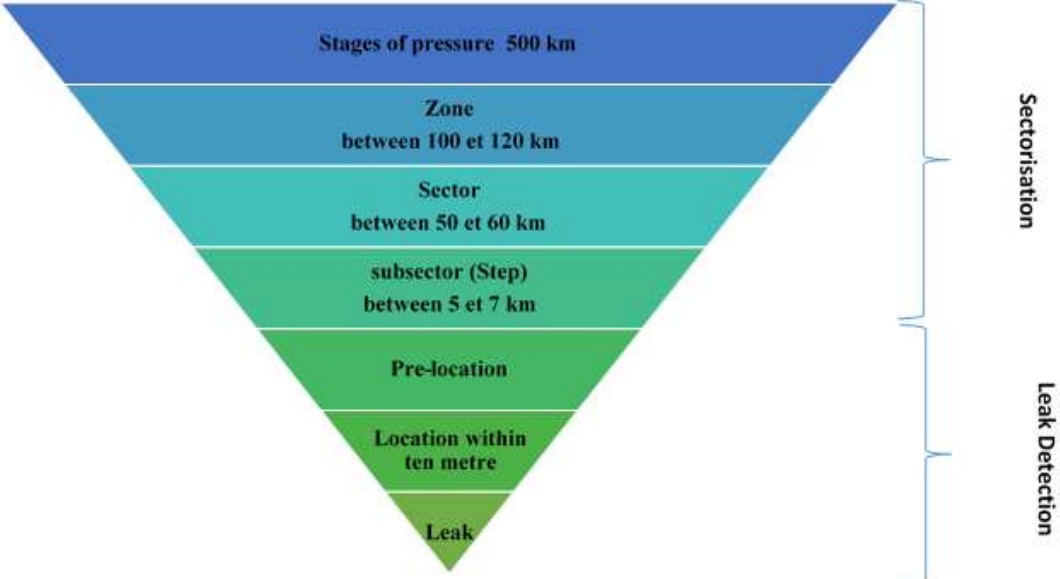

**Figure 10 Stages of sectorization**

The objective of sectorization is to define priorities between different sectors and to estimate or even quantify the level of leakage. It defines fugitive areas larger than the linear kilometer;

The objective of the pre-location is to check the presence of leaks in a given sector and to determine their position with a precision of the order of magnitude of the hundred meters. The correlation is sometimes used to confirm leak position. It consists in positioning 2 sensors on access points of the network (if possible on both sides of the leak) and to seek the similarities between the noises which they record.When a leak noise is identified, it is possible to calculate its position, knowing:

- the distance between the two sensors.

The objective of this correlation is to define the position of a leak with a precision of the order of one meter, to avoid extra cost of earth work without leak.

**The objective of this research was to skip some of the steps, especially all the steps of sectorization, presenting a high day operation cost of the network and achieve directly the last step of prelocalization. The purpose was to pre localise a leak within a hundred meters radius2.3.2 Pre-localization of leaks by learning data**

**-** Random Forest

The type of learning we apply to anomaly detection in this article is a supervised learning. In Zhang et al., (2008), decision tree forests are used to detect intrusions from the network.

To execute the anomaly detection method by supervised learning, we used the statistical software -R (Zhang et al. 2008).

Random Forest is a type of tree based supervised learning algorithm (Ho, 1995). It uses many decision trees to aggregate the answer. In this paper, the supervised Random Forest algorithm was used as technique to detect the leaks (Breiman, 2001). In addition to its efficiency, this algorithm is famous for its ability to treat big-data.

The random forest optimization principle is based on the combination of multiple decision trees, to extract different classes from the original raw dataset. Then, the average classes are determined based on the classes outputted by the decision trees used. Thus, the performance of the resulting model is enhanced, compared to one decision tree model, and the ability to apply the resulting model in other datasets is acquired. Figure 11 illustrates the principle of running a random forest algorithm.

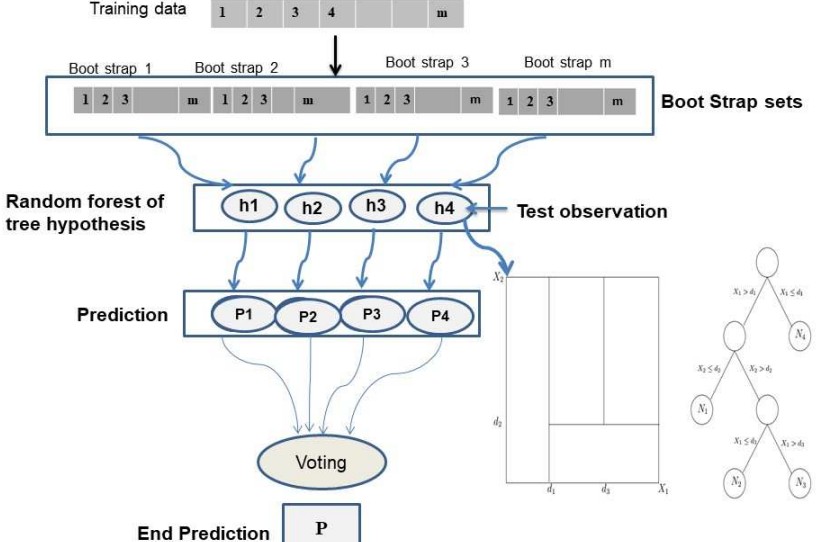


**Figure 11 Random Forest Tree**

- Leakage Detection Algorithm:

The input data are: Training data td, (sensor node number, the distance between simulation node and the sensor, the leakage flow Emitter Coefficient and Pressure at the sensors).

The process engaged in the proposed leak detection is concisely discussed in the following Algorithm:

Algorithm:

1: Start {

2: Load Training data network parameters

3: Read network parameters and initialise

4: for node i= 1 to nt, (nt: The number of nodes in the network)

Run hydraulic analysis and compute leakage Qleak

5: if Qleak < tolerance (relatively low)

6: Print "L= 0" No leaking node"

7: else

8: i: Print "Leaking node ID"

9: if Qleak > tolerance

10: Print "L= 1" leaking node"

11: else

12: i: Print "Leaking node ID"

13: Display "Distance from sensor, pressure node, Qleak ..."

14: end if

15: end if

16: end for i

17: Stop}

## 2.4 Preparation of input data for the algorithm

Several pressure profiles for reference (without leaks) are required to attain satisfactory level of prediction from the data analysis algorithms. This pressure profiles are obtained by using EPANET. Around 7 references cases are added to the table 1. A first case is the pressure reference that has been simulated, the others constitute a translation of the reference curve of +0.1, +0.2, +0.3, -0.1, -0.2, and 0.2 meters, forming envelope with 0.6m amplitude as shown in Fig. 12 below case of P14 measuring point.

we imposed this error range to avoid false results. According to the field tests we carried out, a leak of 6 l/s implies a minimum pressure drop of 0.3 m. the band is a translation of +/- 0.3m, those 7 references cases are added to pressure profiles without leak.

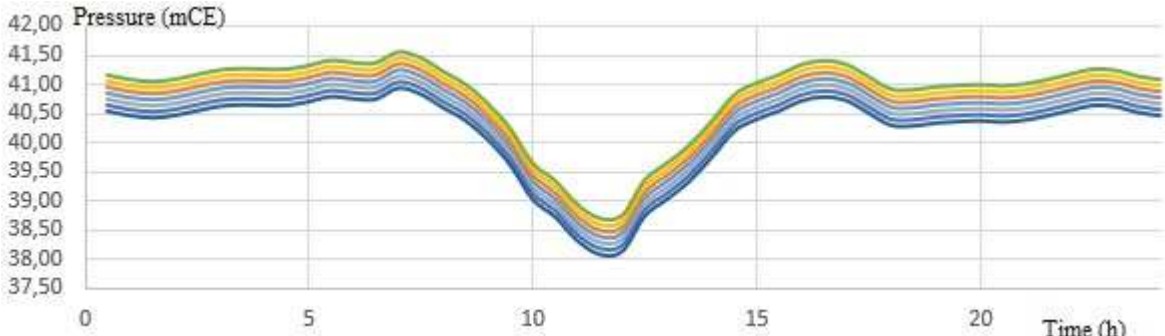

**Figure 12 The reference pressures profile at P14**

For each measurement point we considered seven reference curves. This envelope is the area of variation of the daily pressure, it was calculated from several measurements of pressure spread over several days.

To simulate the artificial leaks in the study area, the leakage rate was created by opening fire hydrants on three locations at three times (during night-time hours, peak time and during off-peak periods of the day) with a leak size of 6 l/s and 17 l/s.
During the leaks simulation, the results are recorded and localized with four pressure measurement sensors placed on fire hydrants.

We then simulated these leaks on EPANET model by taking stock of the time interval of real simulations, taking a time step of 5 min equal to the recording step of the sensors in the field.

A total of 4 leaks were simulated by opening fire hydrants rated A, B and C, the artificial leaks locations are shown in Fig. 13.

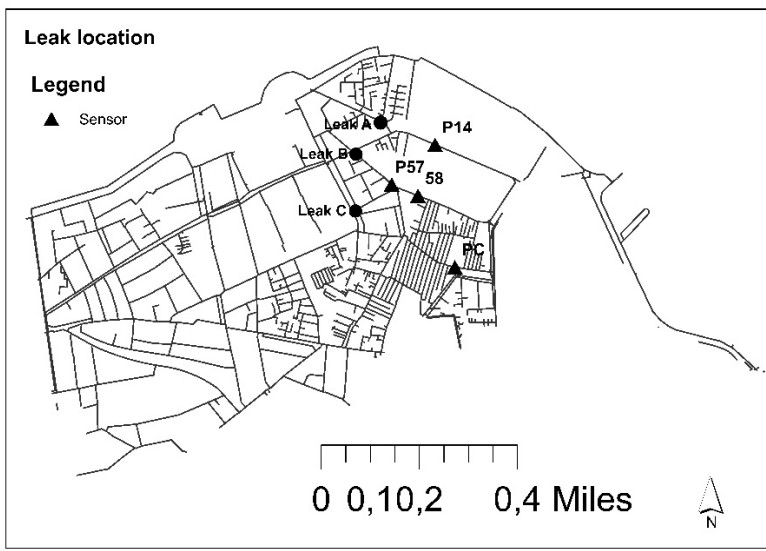


**Figure 13 Sensors and leaks locations**

The leak localization method depends on the head loss, itself depends, among other factors, on the size of the leak. Thus, if we have a limited leak, the sensors could not detect the small head loss, a leak of 6 l/s is chosen as a lower simulation limit. As upper limit, we chosen the 17 l/s leak, and we presume that those bypassing this limit finish by surfacing, and therefore do not
require any localization process. (Pérez et al., 2014a).

At "PI A" we simulated two leaks of 6 l/s and 17 l/s; for the "PI B" a small leak of 6 l/s and for "PI C" a leak of 17 l/s (Table 3).

The flow rate at each fire hydrant was controlled by using a pressure measurement and flowmeter for fire post (PFP). The hydrants were kept opened for around 30 minutes to collect data. The time of simulation was limited to 30 minutes for reasons
of water conservation and safety considerations.

**Table 3 Leaks information in the study area**

| ID of hydrants | Emitter Coefficient | leak flow rate at 04:00 am | Duration of simulation (min) | leak flow rate at 11:16 am | Duration of simulation (min) | leak flow rate at 05:00 pm | Duration of simulation (min) |
|---|---|---|---|---|---|---|---|
| PI A | 0.8 | 6 L/s | 30 | 6 L/s | | 6 L/s | |
| PI A | 3.0 | 17 L/s | 20 | 17 L/s | 30 | 17 L/s | 30 |
| PI B | 0.8 | 6 L/s | 20 | 6 L/s | | 6 L/s | |
| PI C | 3.0 | 17 L/s | 30 | 17 L/s | | 17 L/s | |

## 3. Results and Discussion

**3.1 Data-reading pressure at sensor**

The performance of the pressure for the artificial leak simulation during May the 3rd is shown in Fig. 9. The blue line in Fig.14 shows the daily pattern of pressure at P14.

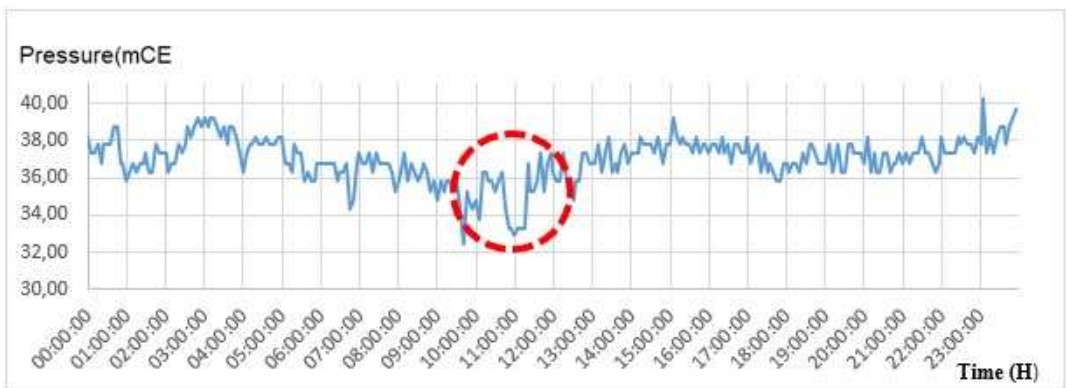

**Figure 14 Pressure profile of 3/5/17 at P14**

The red circle shows the decrease of the pressure during the real simulation.

These leaks were simulated on the EPANET model by choosing the same time slot of the real simulations.

The model was calibrated without errors. It was tested and validated by comparing the measured and simulated pressures.

The results are shown in Fig. 15 and 16.

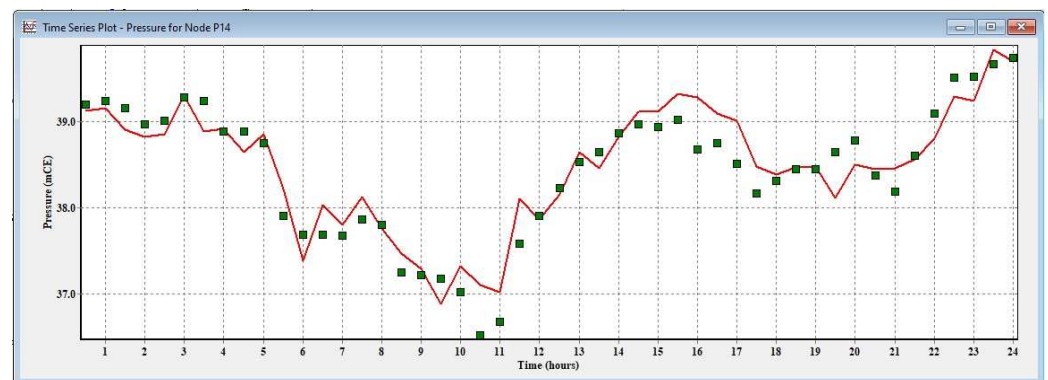

**Figure 15 Pressure profile at P14**

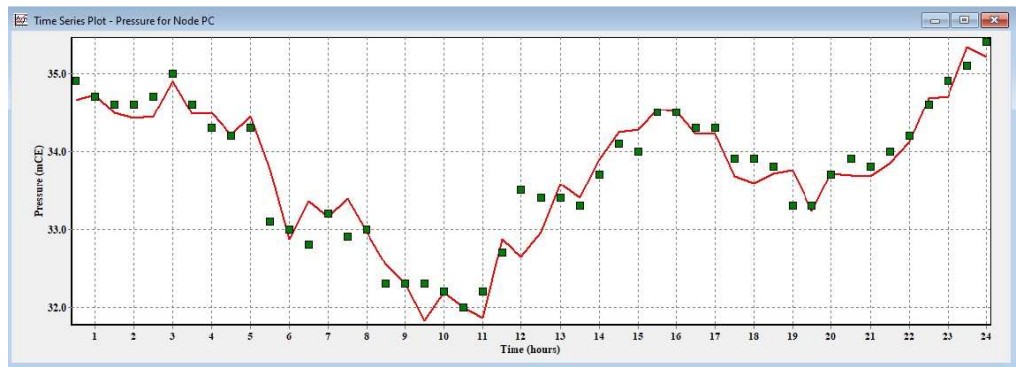

**Figure 16 Pressure profile at PC**

The figures show an example of comparison between modeled pressure and measured pressure in the study area at sensor P14 and PC. The red line is the simulated pressures in Epanet and the green squares is the measured pressure.

The comparison shows that the results of the simulation are very close to what is measured, the model is calibrated and is ready for simulation.

The calibration is also made against the flow at the three inputs of the study area, the result of the flow simulation (red line) are very close to what is measured green squares as shown in Fig.17.

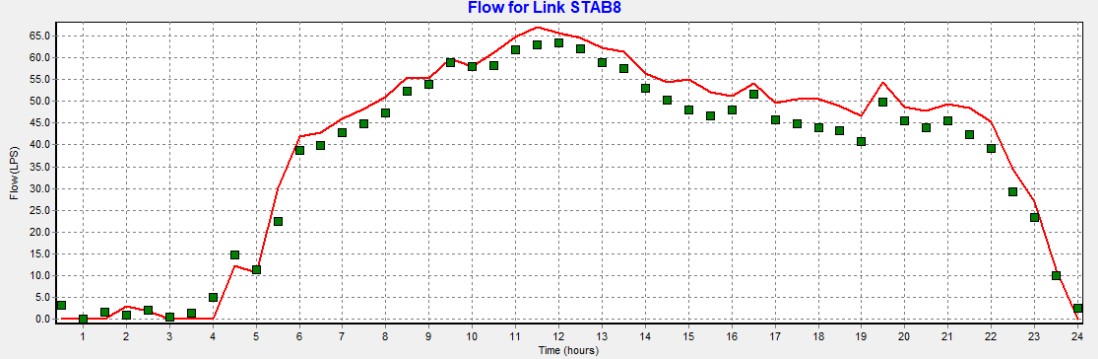

**Figure 17 flowrate calibration**

To calculate the magnitude of the leak for each time and node, Eq.2 was used. The following Table shows the emitter coefficients simulated on each network node (Sebbagh et al., 2017).

**Table 4 Emitter coefficients corresponding to simulated leaks**

| ID Leak | Flow | Emitter coefficient |
|---------|------|---------------------|
| LA-1 | 6 l/s | 1 |
| LA-2 | 17 l/s | 2,2 |
| LB | 6 l/s | 1 |
| LC | 17 l/s | 2,4 |

Table 5 resumes where the leaks are spatially spotted and the value of leakage flow at each node of the network, using the Random Forest algorithm.

**Table 5 Quantification of the leaks.**

| Date | L | Emitter coefficient | Flow rate | D1 | D2 | D3 | D4 |
|------|---|---------------------|-----------|-----|-----|-----|-----|
| 22/04 | 0 | 0,41 | 2,47 | 91 | 45 | 39 | 84 |
| 23/04 | 1 | 2,21 | 13,77 | 529 | 362 | 336 | 358 |
| 24/04 | 0 | 0,33 | 2,62 | 122 | 47 | 41 | 86 |
| 25/04 | 0 | 0,34 | 2,31 | 48 | 53 | 44 | 148 |
| 26/04 | 1 | 2,99 | 17,89 | 525 | 344 | 320 | 332 |
| 27/04 | 0 | 0,32 | 2,10 | 65 | 37 | 46 | 45 |
| 28/04 | 0 | 0,01 | 0,04 | 4 | 0 | 0 | 2 |
| 09/05 | 0 | 0,01 | 0,03 | 3 | 0 | 0 | 2 |
| 10/05 | 0 | 0,01 | 0,03 | 3 | 0 | 0 | 2 |
| 11/05 | 0 | 0,03 | 0,27 | 37 | 16 | 14 | 18 |
| Leak LA-1 | 1 | 2,35 | 14,12 | 229 | 239 | 291 | 591 |
| Leak LA-2 | 1 | 2,60 | 16,04 | 201 | 214 | 259 | 517 |
| Leak LB | 1 | 2,62 | 16,25 | 253 | 207 | 271 | 464 |
| Leak LC | 1 | 3,00 | 18,34 | 288 | 110 | 164 | 345 |

Note that for the 4 last leaks we have L = 1 which means that the algorithm classified them as leaks. However, the leak is over-estimated. The algorithm shows the days of 23/04 and 26/04 as being cases of leaks.

The results of these two days are not used because the change in pressure profile during these days was not because of a leak but because there was closure of some valves in the area to do some work,

For other days we have indeed L = 0

The data analysis confirms that, for an emission coefficient of at least 2, the leaks pre-localization via the adopted method is possible in particularly for flows passing 10 l/s. Indeed, these values of flow provoke important "head loss" easily detected by the pressure sensors implemented within the acting zone, which confirms the hypothesis made at the beginning of this study.

Two phenomena can explain the limits of the current method in terms of its capacity to detect leaks with low flow values. First of all, according to Jarrige et al., (2011) several factors may influence the leak noise propagation til the sensors, such as: the material type, the pipe diameter, and more importantly the pipe roughness. In fact, the misevaluation of this last factor influences the reference pressure calculation. According to Paquin et al., (2000) the results prove that it is necessary to measure the real roughness in order to interpret correctly a simulated and a measured pressure.

The second reason beyond the limited performance of the proposed method to spot leak characterized by low flow was highlighted by a study of Mirats-Tur et al., (2014). In their paper the authors demonstrated that a mis-calibrated hydraulic model (in terms of the topographic structure and its parameters), the precision regarding the estimation of the spatial water need distribution within the sector of application and the precision of the sensors embedded in the network are all eventual causes of the mentioned detection issues.

In fact, like the Mirats-Tur et al. (2014) study, our proposed method is mainly relying on the sensitivity analysis of the pressure measures to the water demand fluctuation in the network nodes. The only difference is that we simulated leak flow series at the level of fire hydrant according to different buffer (radius of 100m, 200m, 300m..), and the only aim is to study the influence radius of each sensor to detect the real flow simulated. This analysis permitted an optimal distribution of sensors in the network. For Mirats-Tur et al (2014), although the sensors were narrowly installed, the leak position was determined within a 150 m

radius, compared to the 100 m radius of our method.

The presented results are outputted from a model established using measures from the network. Some of these measures are considered to have a good precision, and other have a certain level of uncertainty. For instance, the roughness and the nodes' elevations measures are highly impacted by uncertainties. Another factor that impact the precision of the proposed method is the measuring devices in terms of their recording interval (the pressure is measured each 5 minutes). In order to optimize the

detection, and to focus on the leaks with high head losses spotted by standard sensors, it is recommended to use sensors with high frequency, capable of recording a high number of samples. This will help detect the small pressure variation caused by low leak flow.

### 3.2 Displaying results

The proposed method outputs are not always reliable. In fact, instead of a deterministic mapping of the leaks, there is a

probabilistic output that maps the probability of occurrence of leaks in space (Pérez et al., 2014b).

If the forecast indicates a leak, which is estimated, to locate it on the map, we have 4 distances from the 4 pressure sensors, around each sensor we draw a circle of radius corresponds to the given distance by the prediction Fig. 18, the ideal would then be that these four circles intersect at a single point which corresponds to the leak point,

The intersection of these 4 circles will then be at the maximum at 12 points if all the circles intersect with each other at two

points, considering two circles, there are three cases:

- The circles intersect at two points, one corresponds to the leak point and the other is his symmetrical with respect to the line passing through the centers of the two circles,

- Circles tangent to each other intersect at a single point that corresponds to the leak point

- The circles do not cross, but if the forecast is good, they can get closer in the leaking zone

By drawing all the points of intersection, around each of them, the leakage location is identified within a 100 m radius.

A good performance with highest probability of having the location of leak when there is a big agglomeration of circle that has a more intersect point. This agglomeration corresponds to the cumulative probability of the given nodes to experience a leakage (Fig. 19).

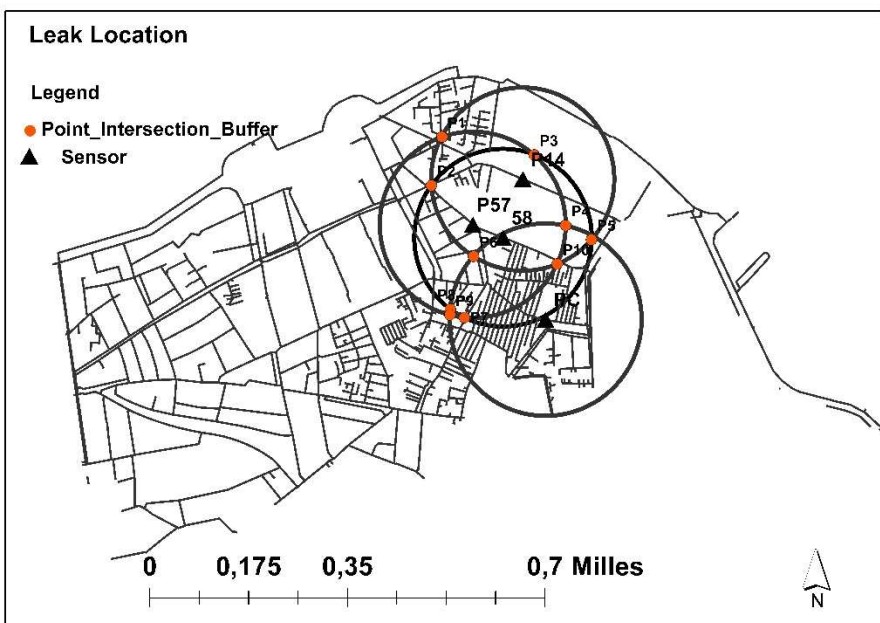

**Figure 18 The spatial intersection of circles**

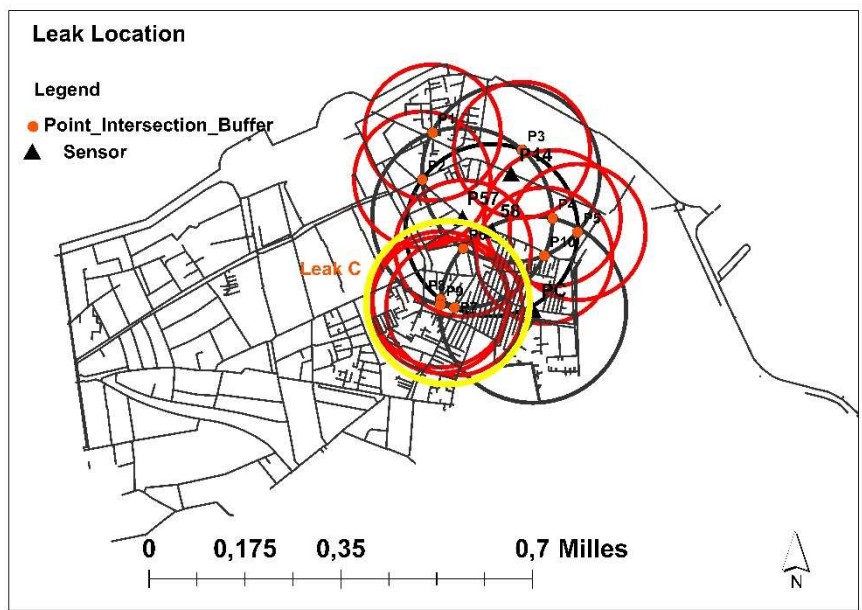

**Figure 19 Spatial location of the leak LC**

Our research objective was the purpose of discovery appropriate solution for detection and localization of leakages and estimation of the size of leakages for a water distribution system.

The results obtained using this approach is satisfying. The leak is identified within a 100 m radius.

That said, the detection of a leak is extremely related to its location within the network. For instance, one located in the looped section of the network is less likely to be spotted in night time. In the mesh part of the network the pressure fallen at the sensor levels are too low, which could lead to disturb by uncertainties in the model, the measured pressures obviously involve significant errors, which reduces, in the analysis, the possibility of detecting leaks of lesser importance.

**4 Conclusions and perspectives**

Our research objective was discovering appropriate solution for detection and localization of leakages and estimation of the size of leakages for a water distribution system. The FAVAD parameters were optimized via a prediction algorithm, to constitute the core of our adopted procedure. The adopted approach necessitates a coupled hydraulic-GIS interface by mean of the random forest algorithm.

This work helped to spot critical leaking points, and therefore contribute in the effort of physical loss reduction. Although, the detection results were not always accurate in term of space localization, the radius of search is reduced substantially, which make the detection rates during field campaigns more successful and less time-consuming.

Conflicts of interests: the authors declare that they have no conflict of interest.

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
