# Peer review of "Prelocalization and Leak detection in drinking water distribution network using modeling-based algorithms: Case study: The city of Casablanca (Morocco)"

_Drinking Water Engineering and Science, 2020_

## Referee Comment (RC1) · Anonymous Referee #1 · 15 Apr 2020

1. Line 10: "water drinking distribution network (WDDN)" ??? Should be "drinking water distribution network (DWDN)" 2. Line 12: "on the field" must be "in the field" 3. Line 25: don't use CC for Climate Change 4. Line 39: don't use capitol after comma 5. Line 41: "DHS" must be "DHS'" 6. Line 61: So there is no flow meter at each inlet?? 7. Line 81. End with " 8. Line 96. What is N 9. Line 119-120. No, this is not dynamic simulation; this is consecutive static simulations. Epanet can only do static calculation, no dynamic calculation including dynamic aspects of accelerating and slowing down of water 10. Line 130 and 139. I don't understand. How can you vary

the Emitter Coefficient? Do you suggest that you keep pressure constant and leak size constant, and let Emitter Coefficient determine the leak flow? Why did you choose this method? How do you model a leak in Epanet? 11. Line 144. Why 1972 simulations? 12. Line 144. Did you also simulate leaks at different times? E.g. at 6 am; at 3 pm; et cetera. 13. Line 172. This seems very precise. How large is the area the you researched? 14. Line 187. Change "number of sensor node" to "sensor node number" 15. Line 210-2015. Why is this? Why simple linear band? 16. Line 225. What is the average demand in the area? Can you show a graph of an average day of measured water demand? 17. Line 225. For how long was de leak simulated? 15 minutes? 1 hours? ?? 18. Line 240. What is simulated (red line?) and what is measured (green squares?)? Please explain

Please also note the supplement to this comment:
https://www.drink-water-eng-sci-discuss.net/dwes-2020-3/dwes-2020-3-RC1-supplement.pdf

---

## Referee Comment (RC2) · Said Rhouzlane (Referee) · 22 Apr 2020

In this paper, the authors investigate numerically, using EPANET code, the leaks in the water distribution network (WDN). The old part of Casablanca city (Morocco) is used as a case of study. The results obtained are compared with those of real simulation of artificial leaks caused by the opening and closing of hydrants. On the whole, the idea behind this study is interesting and promising. So, I support the paper for publication after a revision to improve its scientific content.

[Figure]

- Line 61 you have mentioned "Concerning the instrumentation, the network flow and pressure are monitored through flowmeter" the study area is it moduled, micro-moduled, how many critical points, measurement points ?? all this is not clear in your article - line 63 Âń The average age is 40 years, increasing vulnerability and promoting leaks Âż you have to descreibe more scientificaly. You have to write and talk to selection criteria: (you need to talk about the performance indicators for this area, MNF, ILP for example) - Line 110 "Roughness coefficient of materials" Physical modeling of your network requires the roughness of the pipes. It should be noted that there is great uncertainty about the values considered. How did you minimize these uncertainties and their impact on the results given by EPANET. - Line 119 , 120 you have mentioned ". Dynamic simulation is used to describe the operation of the network during a given period, while taking into account the variation in customers' consumption over time." Physical modeling of your network requires consumption at nodes. It is important to indicate how you estimated this parameter. It should be noted that there is great uncertainty about the nodal distribution of the study area. How did you develop your consumption patterns for dynamic modeling? - Line 135 you have mentioned that you have used the Matlab-software, For EPANET code, please find a better reference - Line 240 you have mentioned "The model was calibrated without errors" What are the calibration data for your model? Discuss them clearly. - line 241 the calibration results of pressure are presented, you need to complete with the flow calibration.

Please also note the supplement to this comment:
https://www.drink-water-eng-sci-discuss.net/dwes-2020-3/dwes-2020-3-RC2-supplement.pdf

---

## Author Comment (AC1) · 23 Apr 2020

Responses to Reviewer 1: On behalf of the authors, I would like to thank the Editor-In-Chief and the reviewer 1, The comments are the addresses below. They will be incorporated into the revised version of the manuscript. Line 10: "water drinking distribution network (WDDN)"??? Should be "drinking water distribution network (DWDN)". This remark is integrated in the revised version of the manuscript (line 10,13,23,46,and 53)

[Figure]

Line 12: "on the field" must be "in the field This remark is integrated in the revised version of the manuscript ( line 12)

Line 25: don't use CC for Climate Change This remark is integrated in the revised version of the manuscript (line 25, 27,28 and 30)

Line 39: don't use capitol after comma This remark is integrated in the revised version of the manuscript

Line 41: "DHS" must be "DHS'" This remark is integrated in the revised version of the manuscript

Line 61: So there is no flow meter at each inlet?? Each of the three inlet of the zone has its own flow meter in diameter 300, and the hydraulic model was calibrated at each inlet.

Line 81. End with " This remark is integrated in the revised version of the manuscript.

Line 96. What is N Empirical research has repeatedly shown that the Fixed and Variable Area Discharges (FAVAD) principle, which demonstrates the fact that most discharges from pressurized pipelines vary with pressure to a greater or lesser extent. This concept, via the definition of an exponent N1, defines the relationship between the leakage rate and the pressure in case of pressure modulation. According to (Lambert, 2000) and (Rozental, 2010) the leakage exponent N1 varies from 0.5 for a fixed area in rigid pipes as metallic pipes and 1.5 for a flexible area in plastic pipes as shown in Equation below L1/L0=×(P1/P0)ˆN1 where L1 and P1 are respectively the leakage rate and the average pressure in the DHS during the day, L0 and P0 are respectively the leakage rate and average pressure at the minimum night flow (MNF) time, between 2:00 AM and 4:00 AM.

Line 119-120. No, this is not dynamic simulation; this is consecutive static simulations. Epanet can only do static calculation, no dynamic calculation including dynamic aspects of accelerating and slowing down of water (This response will be incorporated in

the revised version of the manuscript) The authors agree that this aspect needs more explanation . What we would like to mention in this section is that, Epanet predicts the dynamic hydraulic behavior within a drinking water distribution system operating over an extended period of time. the pressure changes due to discharge, pressure calculating and changes according to base demand and daily consumption patterns at each node. Figure 1 bellow shows an example of daily consumption patterns in the study area.

Figure 1 daily consumption patterns in the study area the pressure drop during a peak period of consumption is due to higher consumption patterns in the DHS (The pattern provides multipliers that are applied to the Base Demand to determine actual demand in a given time period).

Line 130 and 139. I don't understand. How can you vary the Emitter Coefficient? Do you suggest that you keep pressure constant and leak size constant, and let Emitter Coefficient determine the leak flow? Why did you choose this method? How do you model a leak in Epanet? (This response will be incorporated in the revised version of the manuscript)

There are two ways to model a water leak in a hydraulic network model EPANET (i) as an additional nodal demand, and (ii) by adding a leak valve to each node Fig 2a and 2b.

Figure 2 Ways to model a water leak in a hydraulic network model where QDi is the node base demand (consumption), QADi is the node additional demand , Pi is the node pressure, Ci is the leak valve coefficient and Qi is the leakage flow. According to Cobacho et al., 2014 the best way to represent leakage in a hydraulic network model is not by means of an additional demand, but rather by adding a leak valve to each node, the dynamic behavior of leakage is appreciated in this case rather than the case as an additional demand. In Epanet, the closest element to a leak valve is the emitter, which presents an open valve to the atmosphere, the emitter behaviour equation is shown in

equation below: "Qleak j = C x Pj" ^N where Qleak is the leakage flow rate at node j, Pj is the pressure, C is the emitter coefficient, and N is the pressure exponent. The emitter coefficient is placed at the junctions, 4 emitter coefficients ( 0.8, 1.6, 2.4 and 3.2) were used in this simulation step at each node, which correspond respectively to approximate flow leaks: 5, 10, 15 and 20 l/s, the objective of this simulation step is to vary the leak rate at nodes, to calculate and generate the new profile of pressures at each changes in flow rate. The simulation results obtained constitute the training data for our machine learning model. But before launching the simulations, the model must therefore be calibrated against measured data of pressure and flow rate (as mentioned in line 240).

Line 144. Why 1972 simulations? The purpose of this 1972 simulations was to stimulate the pressure behavior at each node, the area contains 493 nodes, thus, we simulated 4 leaks with emitter coefficients equal to 0.8, 1.6, 2.4 and 3.2. This makes a total of 1972 simulations. Each time, the leakage rate, the emitting coefficient, the location of the leak, the distances from the four sensors P14, P57, P58 and PC are noted, as well as the maximum, minimum and average pressures at these measuring points. the authors would like to record the maximum data of pressure variations in the case even if the leaks are triggered in the looped far part on the network. A completely new base for data learning.

Line 144. Did you also simulate leaks at different times? E.g. at 6 am; at 3 pm; et cetera (This response will be incorporated in the revised version of the manuscript)

To simulate the artificial leaks in the study area, the leakage rate was created by opening fire hydrants on three locations at three times (during night-time hours, peak time and during off-peak periods of the day) with a leak size of 6 l/s and 17 l/s. The Table below illustrate the leakage information. Table 1 Leaks information in the study area

During the leaks simulation, the results are recorded and localized with four pressure measurement sensors placed on fire hydrants. We then simulated these leaks on

EPANET model by taking stock of the time interval of real simulations, taking a time step of 5 min equal to the recording step of the sensors in the field.

Line 172. This seems very precise. How large is the area the you researched? This response will be incorporated in the revised version of the manuscript to answer this question, we would like to remind the existing techniques of leak detection. the following figure 3 summarizes the existing pre-localization stages, these stages make it possible to go from hundreds of km of network to tens of m, by proceeding by elimination: In the identified leaky zones, the night flow of sectors is measured then sub-sectors to identify leaking sections and precisely guide acoustic detection and then the location of leaks. this system presents a high operation cost and high equipment cost for leak detection.

Figure 3 Stages of sectorization In the line 172 we have quoted "The objective of the location is to define the position of a leak with a precision of the order of one meter". in comparison with existing techniques like acoustic correlation, The correlation consists in positioning 2 sensors on access points of the network (if possible on both sides of the leak) and to seek the similarities between the noises which they record. When a leak noise is identified, it is possible to calculate its position, knowing: - the distance between the two sensors. The objective of this correlation is to define the position of a leak with a precision of the order of one meter, to avoid extra cost of earth work without leak. in view of the cost of this approach mentioned above, we worked on this approach of virtual leaks research, without the closure the valves in the field, without sectorization, only through a well-calibrated hydraulic model, the data of pressure due to the conditions of the leakage experiments, are compared to the pressure of the previous day without leakage, this data will be associated with an algorithm that predict the location of the leak. Our goal was to skip some stages, especially all the steps of sectorization, presenting a high day operation cost of the network and achieve directly the last step of prelocalization. The purpose was to prelocalise a leak within a 100 to 150m radius

Line 187. Change "number of sensor node" to "sensor node number" This remark is integrated in the revised version of the manuscript.

Line 210-2015. Why is this? Why simple linear band? we imposed this error range to avoid false results. According to the field tests we carried out, a leak of 6 l/s implies a minimum pressure drop of 0.3 m. the band is a translation of +/- 0.3m, those 7 references cases are added to pressure profiles without leak.

Line 225. What is the average demand in the area? Can you show a graph of an average day of measured water demand? the following figure 4 illustrates the system flow balance in this study area: (This response will be incorporated in the revised version of the manuscript)

Figure 4 curve of the average demand in the study area Line 225. For how long was de leak simulated? 15 minutes? 1 hours? ?? (This response will be incorporated in the revised version of the manuscript)

As I mentioned above, the flow rate at each fire hydrant was controlled by a CPI for flow rate and pressure tests on fire hydrants. The hydrants were kept opened for around 30 minutes to collect data. The time of simulation was limited to 30 minutes for reasons of water conservation and safety considerations.

Line 240. What is simulated (red line?) and what is measured (green squares?)? Please explain (This response will be incorporated in the revised version of the manuscript)

In line 240 the Fig. 10 shows an example of comparison between modeled pressure and measured pressure in the study area at sensor P14. The red line is the simulated pressures in Epanet and the green squares is the measured pressure. The comparison shows that the results of the simulation are very close to what is measured, the model is calibrated and is ready for simulation.

New Reference: Lambert, A.: What do we know about pressure-leakage relationships

in distribution systems. In Proceedings of the IWA Conference in Systems Approach to Leakage Control and Water Distribution System Management, Brno, Czech Republic, 16–18 May 2001

Please also note the supplement to this comment:
https://www.drink-water-eng-sci-discuss.net/dwes-2020-3/dwes-2020-3-AC1-supplement.pdf

[Figure]

**Fig. 1.**

[Figure]

**Fig. 2.**

Stage of pressure around 500 Km

Zone 100 Km

Sector 30-50 km

Step 10 km

Prelocalization & leak detection
through acoustic detection

Localization
about ten m

Leak

Sectorization

Leak Detection

**Fig. 3.**

[Figure]

**Fig. 4.**

| ID of hydrants | Emitter Coefficient | leak flow rate at 04:00 am | Duration of simulation (min) | leak flow rate at 11:16 am | Duration of simulation (min) | leak flow rate at 05:00 pm | Duration of simulation (min) |
|---|---|---|---|---|---|---|---|
| PI A | 0.8 | 6 L/s | 30 | 6 L/s | | 6 L/s | |
| PI A | 3.0 | 17 L/s | 20 | 17 L/s | 30 | 17 L/s | 30 |
| PI B | 0.8 | 6 L/s | 20 | 6 L/s | | 6 L/s | |
| PI C | 3.0 | 17 L/s | 30 | 17 L/s | | 17 L/s | |

**Fig. 5.**

---

## Referee Comment (RC3) · Said Rhouzlane (Referee) · 30 Apr 2020

Mr. Taghlabi answered all my questions.

---

## Author Comment (AC2) · 30 Apr 2020

On behalf of the authors, I would like to thank the Editor-In-Chief and the reviewer Pr Said RHOUZLANE The comments are addressed below. They will be incorporated into the revised version of the manuscript. Line 61 you have mentioned "Concerning the instrumentation, the network flow and pressure are monitored through flowmeter" the study area is it moduled, micro-moduled, how many critical points, measurement points ?? all this is not clear in your article

[Figure]

The study area is a very low area which is at an average NGM level of 6 m above sea level, it is supplied by a semi buried reservoir which is at the NGM 85 m level, with 4 m in height, around 88 hydraulic head., which generates significant pressure at night (outside normal hours of consumption), the sector also experiences a significant night flow and a high frequency of recurrence of leaks for these three reasons, the study area was modulated, but this first modulation has not halted the reappearance of leaks, this is why, it underwent a second pressure modulation in order to reduce the breakage rate, in indeed the area today is micro modulated. The sector contains a single critical pressure point that is continuously monitored.

line 63 Âń The average age is 40 years, increasing vulnerability and promoting leaks Âż you have to descreibe more scientifficaly. You have to write and talk to selection criteria: (you need to talk about the performance indicators for this area, MNF, ILP for example)

The study area of about 42 km has a significant minimum night flow around 40 LPS between 2:00 and 4:00 in the morning, which implies a high probability of the presence of physical losses as illustrated in Figure 1 below .

Figure 1 MNF in the study area

The study area has a very high linear loss index (ILP) of around 54 m3 / d / km, which implies a very poor network condition.

Line 110 "Roughness coefficient of materials" Physical modeling of your network requires the roughness of the pipes. It should be noted that there is great uncertainty about the values considered. How did you minimize these uncertainties and their impact on the results given by EPANET. Different factors are attributable to the uncertainty of the pressure measured on the network, the most important of which is the fluctuation of the pressure in the network. According to the pressure differences observed on the ground when determining roughness, according to the work of Paquin et al., (2000) this error is estimated at 0.3 mCE. Regarding the error in modeling the reference pressures,

a value of 0.2 mCE is considered realistic. The total uncertainty related to the pressure difference between the measured pressure and the reference pressure is therefore 0.5 mCE. The results which have been presented come from a model established from data collected in the field and at the level of a pipe section. This data is believed to be accurate or with great precision, while the rest of the network has some level of uncertainty. This is the case for the roughness of the pipes, the elevations of the nodes and the consumption of the residences. Various factors including the measuring devices and the state of equilibrium of the network can influence the accuracy of the value of the roughnesses measured in the field, as well as for the estimation of roughness for the other conduits of the network. The following figure presents 3 curves which correspond to the pressure drop caused by a leak in the middle of a pipe without a junction, one with roughness as measured in the field, and two others with limit values taking into account a +/- 10% error related to this parameter (ÔŘ + 10%, ÔŘ −10%).

Figure 2 Influence of the error on the roughness of the pipe

When the roughness is increased, the pressure difference caused by a leak is greater. Conversely, when the pipes are smoother (plastic), the pressure difference decreases. Leak detection by the method studied therefore seems more promising on networks whose hydraulic capacity is weakened by corrosion rates.

Line 119 , 120 you have mentioned ". Dynamic simulation is used to describe the operation of the network during a given period, while considering the variation in customers' consumption over time." Physical modeling of your network requires consumption at nodes. It is important to indicate how you estimated this parameter. It should be noted that there is great uncertainty about the nodal distribution of the study area. How did you develop your consumption patterns for dynamic modeling?

To calculate demand, we base on the billing sectors. The billing sectors correspond to former consumer sectors, and not to the current sectorization. The following figure shows the succession sectors of the study sector.

Figure 3 Billing sectors intersecting with the study area

And even though they do not coincide with the hydraulic sectors specific to the sectoring, we proceed to the calculation of the intersections of the surfaces of these billing sectors with the perimeter to be evaluated to determine their reference volumes Eq (1):

Q_moy (secteur 1)=$\sum (S\_iS\_(secteur1))/S\_i*Q\_(moyi) Eq(1)$

The average flow of each billing sector will then be divided by the number of nodes in this sector and the value will be assigned to all of them so as to have a spatial distribution of consumption close to reality.

• The consumption pattern: This is the daily consumption profile at all the nodes in our sector, we build it from the total flow of our sector imported from the network manager database. The following graph represents the raw signal of the measurement: We proceeded to smooth the curve by moving average of period 7 in order to eliminate small variations and we reported the flows in time steps 30 min then we plotted the daily average consumption of the sector. The multiplying coefficients are calculated by dividing the hourly flows by the daily average in Figures 4 and 5.

Figure 4 flow in the study area with calculation of the average

Figure 5 Curve of pattern demand

Line 135 you have mentioned that you have used the Matlab-software, For EPANET code, please find a better reference For EPANET code, the original work based on: D.G. Eliades, M. Kyriakou, S. Vrachimis and M.M. Polycarpou, "EPANET-MATLAB Toolkit: An Open-Source Software for Interfacing EPANET with MATLAB", in Proc. 14th International Conference on Computing and Control for the Water Industry (CCWI), The Netherlands, Nov 2016, p.8. (doi:10.5281/zenodo.831493).

Line 240 you have mentioned "The model was calibrated without errors" What are the calibration data for your model? Discuss them clearly In the report, I have mentioned just the pressure calibrating, the calibration is also made against the flow at the three

inputs of the study area, below the correlation curves of the two parameters figure 6 and 7.

Figure 6 Correlation for flow

Figure 7 correlation for pressure

Line line 241 the calibration results of pressure are presented, you need to complete with the flow calibration

the figure below shows the average demand curve at one of the three inlets.

Figure 8 flow rate calibration

Please also note the supplement to this comment: https://www.drink-water-eng-sci-discuss.net/dwes-2020-3/dwes-2020-3-AC2-supplement.pdf

[Figure]

**Fig. 1.**

[Figure]

**Fig. 2.**

[Figure]

**Fig. 3.**

[Figure]

**Fig. 4.**

[Figure]

| Période | 1 | 2 | 3 | 4 | 5 | 6 | 7 | 8 |
|---|---|---|---|---|---|---|---|---|
| Multiplicateur | 0.547 | 0.497 | 0.467 | 0.443 | 0.42 | 0.397 | 0.397 | 0.3 |

**Fig. 5.**

[Figure]

**Fig. 6.**

[Figure]

**Fig. 7.**

[Figure]

**Fig. 8.**

---

## Author Comment (AC3) · 15 May 2020

Thank you for your response, I would be available for any further request

---

## Author Comment (AC4) · 15 May 2020

I would like to add a comment on this question: (3) Line 110 "Roughness coefficient of materials" Physical modeling of your network requires the roughness of the pipes. It should be noted that there is great uncertainty about the values considered. How did you minimize these uncertainties and their impact on the results given by EPANET.

The pressure and flow within a network vary depending on the time of day. Because of reduced night water consumption, the average pressure inside the network increases.

Based on the values established for these two parameters, three simulations were carried out to analyze the influence of the measurement period on the calculated pressure difference. The following figure shows three pressure drop curves caused by a leak halfway through a pipe without a junction but with 3 base flow rates flowing in the different pipe: 5, 10 and 15 L / s.

Thus, for the same leakage flow rate, it is noted that the pressure drop is all the greater the higher the base flow rate flowing in the pipe, this is due to the power 2 of the flow rate in the Darcy-Weisbach formula. Pressure drops are then greater during periods of high consumption, even for low leakage rates.

Figure 1 Pressure drops vs Leakage rate

———————————————————————

[Figure]

**Fig. 1.**

---

## Author Response (AR1)

[revised manuscript text omitted]

**Point-by-point reply to the comments**

**We thank the Editor-In-Chief, reviewer 1 and Pr Said RHOUZLANE for their helpful comments. The comments from reviewers were constructive and helped to improve the quality of the manuscript. We have taken the comments on board to improve and clarify the manuscript. Please find below a detailed point-by-point response to all comments.**

**reviewer 1 comments:**

**Referee comment:** "water drinking distribution network (WDDN)"??? Should be "drinking water distribution network (DWDN)".

Author response: (marked up manuscript lines 10,14,47, and 55)

We have replaced the phrase 'water drinking distribution network (WDDN)' with 'drinking water distribution network (DWDN) (line 140).

**Referee comment:** "on the field" must be "in the field

Author response: (marked up manuscript line 14)

This change was made.

**Referee comment**: don't use CC for Climate Change

Author response: (marked up manuscript lines 26, 28,29 and 31)

We have removed the citation from the manuscript.

**Referee comment**: don't use capitol after comma

Author response: (marked up manuscript line 40)

this remark is integrated in the revised paragraph

**Referee comment**: "DHS" must be "DHS'"

Author response: (marked up manuscript line 40)

This change was made.

Referee comment: So there is no flow meter at each inlet??
Author response: (marked up manuscript lines 64-65)

Each of the three inlet of the zone has its own flow meter in diameter 300, and the hydraulic model was calibrated at each inlet.

Referee comment: End with "
Author response: (marked up manuscript line 93)

This remark is integrated in the revised version of the manuscript.

Referee comment: What is N
Author response: (marked up manuscript lines 97-108,455-457)

Empirical research has repeatedly shown that the Fixed and Variable Area Discharges (FAVAD) principle, which demonstrates the fact that most discharges from pressurized pipelines vary with pressure to a greater or lesser extent. This concept, via the definition of an exponent N1, defines the relationship between the leakage rate and the pressure in case of pressure modulation.

According to (Lambert, 2000) and (Rozental, 2010

$L1/L0 = \times(P1/P0)^{N1}$

where L1 and P1 are respectively the leakage rate and the average pressure in the DHS during the day, L0 and P0 are respectively the leakage rate and average pressure at the minimum night flow (MNF) time, between 2:00 AM and 4:00 AM.

The number of references were increased to 1.

Lambert, A.: What do we know about pressure-leakage relationships in distribution systems. In Proceedings of the IWA Conference in Systems Approach to Leakage Control and Water Distribution System Management, Brno, Czech Republic, 16–18, 2001.

Referee comment: No, this is not dynamic simulation; this is consecutive static simulations. Epanet can only do static calculation, no dynamic calculation including dynamic aspects of accelerating and slowing down of water
Author response: (marked up manuscript lines 135-142)

 It predicts the dynamic hydraulic behavior within a drinking water distribution system operating over an extended period of time. the pressure changes due to discharge, pressure calculating and changes according to base demand and daily consumption patterns at each node. The pressure drop during a peak period of consumption is due to higher consumption patterns in the DHS (The pattern provides multipliers that are applied to the Base Demand to determine actual demand in a given time period).
**Figure 4 daily consumption patterns in the study area**

**Referee comment**: I don't understand. How can you vary the Emitter Coefficient? Do you suggest that you keep pressure constant and leak size constant, and let Emitter Coefficient determine the leak flow? Why did you choose this method? How do you model a leak in Epanet?

Author response: (marked up manuscript lines 175-189)

There are two ways to model a water leak in a hydraulic network model EPANET (i) as an additional nodal demand, and (ii) by adding a leak valve to each node Fig 2a and 2b.

**Figure 5 Ways to model a water leak in a hydraulic network model**

where $QD_i$ is the node base demand (consumption), $QAD_i$ is the node additional demand, $P_i$ is the node pressure, $C_i$ is the leak valve coefficient and $Q_i$ is the leakage flow.

According to Cobacho et al., 2014 the best way to represent leakage in a hydraulic network model is not by means of an additional demand, but rather by adding a leak valve to each node, the dynamic behavior of leakage is appreciated in this case rather than the case as an additional demand.

In Epanet, the closest element to a leak valve is the emitter, which presents an open valve to the atmosphere, the emitter behaviour equation  as shown above in equation 2

$^{}$

The emitter coefficient is placed at the junctions, 4 emitter coefficients ( 0.8, 1.6, 2.4 and 3.2) were used in this simulation step at each node, which correspond respectively to approximate flow leaks: 5, 10, 15 and 20 l/s, the objective of this simulation step is to vary the leak rate at nodes, to calculate and generate the new profile of pressures at each changes in flow rate.

The simulation results obtained constitute the training data for our machine learning model

**Referee comment**: Why 1972 simulations?

Author response: (marked up manuscript lines 201-208)

The purpose of this 1972 simulations was to stimulate the pressure behavior at each node, the area contains 493 nodes, thus, we simulated 4 leaks with emitter coefficients equal to 0.8, 1.6, 2.4 and 3.2. This makes a total of 1972 simulations. Each time, the leakage rate, the emitting coefficient, the location of the leak, the distances from the four sensors P14, P57, P58 and PC are noted, as well as the maximum, minimum and average pressures at these measuring points. the authors would like to record the maximum data of pressure variations in the case even if the leaks are triggered in the looped far part on the network. A completely new base for data learning.

**Referee comment**: Did you also simulate leaks at different times? E.g. at 6 am; at 3 pm; et cetera

Author response: (marked up manuscript lines 298-305)

To simulate the artificial leaks in the study area, the leakage rate was created by opening fire hydrants on three locations at three times (during night-time hours, peak time and during off-peak periods of the day) with a leak size of 6 l/s and 17 l/s.

Table 3 Leaks information in the study area

During the leaks simulation, the results are recorded and localized with four pressure measurement sensors placed on fire hydrants.

We then simulated these leaks on EPANET model by taking stock of the time interval of real simulations, taking a time step of 5 min equal to the recording step of the sensors in the field.

**Referee comment**: This seems very precise. How large is the area the you researched?
Author response: (marked up manuscript lines 228-233, 238-246)

The following figure 10 summarizes the existing pre-localization stages, these stages make it possible to go from hundreds of km of network to tens of m, by proceeding by elimination: In the identified leaky zones, the night flow of sectors is measured then sub-sectors to identify leaking sections and precisely guide acoustic detection and then the location of leaks.

**Figure 8 Stages of sectorization**

The correlation is sometimes used to confirm leak position. It consists in positioning 2 sensors on access points of the network (if possible on both sides of the leak) and to seek the similarities between the noises which they record.When a leak noise is identified, it is possible to calculate its position, knowing:

- the distance between the two sensors.

The objective of this correlation is to define the position of a leak with a precision of the order of one meter, to avoid extra cost of earth work without leak.

~~in view of the cost of this approach mentioned above, we worked on this approach of virtual leaks research, without the closure the valves in the field, without sectorization, only through a well calibrated hydraulic model, the data of pressure due to the conditions of the leakage experiments, are compared to the pressure of the previous day without leakage, this data will be associated with an algorithm that predict the location of the leak.~~

The objective of this research was to skip some of the steps, especially all the steps of sectorization, presenting a high day operation cost of the network and achieve directly the last step of prelocalization. The purpose was to pre localise a leak within a hundred meters radius

**Referee comment**: Change "number of sensor node" to "sensor node number"
Author response: (marked up manuscript line 263)

This change was made

**Referee comment**:  Why is this? Why simple linear band?
Author response: (marked up manuscript line 291-293)

we imposed this error range to avoid false results. According to the field tests we carried out, a leak of 6 l/s implies a minimum pressure drop of 0.3 m. the band is a translation of +/- 0.3m, those 7 references cases are added to pressure profiles without leak

**Referee comment**: What is the average demand in the area? Can you show a graph of an average day of measured water demand?

Author response: (marked up manuscript line 68-70)

615   The following figure 2 shows the daily average of measured water demand

**Figure 2 Daily average water demand of the study area.**

**Referee comment**: For how long was de leak simulated? 15 minutes? 1 hours? ??

Author response: (marked up manuscript line 316-318)

620   the flow rate at each fire hydrant was controlled by using a pressure measurement and flowmeter for fire post  (PFP) . The hydrants were kept opened for around 30 minutes to collect data. The time of simulation was limited to 30 minutes for reasons of water conservation and safety considerations.

**Referee comment**: What is simulated (red line?) and what is measured (green squares?)? Please explain

625   Author response: (marked up manuscript line 337-340)

The Figures show an example of comparison between modeled pressure and measured pressure in the study area at sensor P14 and PC. The red line is the simulated pressures in Epanet and the green squares is the measured pressure.
The comparison shows that the results of the simulation are very close to what is measured, the model is calibrated and is ready for simulation.

630

**The comments from reviewer 2 Pr Said RHOUZLANE were constructive and helped to improve the quality of the manuscript. The comments are addressed below.**

635   **Referee comment**: you have mentioned "Concerning the instrumentation, the network flow and pressure are monitored through flowmeter" the study area is it moduled, micro-moduled, how many critical points, measurement points ?? all this is not clear in your article

Author response: (marked up manuscript line 63)

The study area is a micro modulated sector with a single critical pressure point that is continuously monitored.

640

**Referee comment**: « The average age is 40 years, increasing vulnerability and promoting leaks » you have to descreibe more scientificaly. You have to write and talk to selection criteria: (you need to talk about the performance indicators for this area, MNF, ILP for example)

Author response: (marked up manuscript line 71-72)

645   The figure above shows a significant minimum night flow around 40 LPS between 2:00 and 4:00 AM, with a linear loss index (LLI) of 54 m3/day/km which implies a high probability of the presence of physical losses.

**Referee comment**: "Roughness coefficient of materials" Physical modeling of your network requires the roughness of the pipes. It should be noted that there is great uncertainty about the values considered. How did you minimize these uncertainties and their impact on the results given by EPANET

Author response: (marked up manuscript line 149-163)

Different factors are attributable to the uncertainty of the pressure measured on the network, the most important of which is the fluctuation of the pressure in the network. According to the pressure differences observed on the ground when determining roughness, according to the work of Paquin et al., (2000) this error is estimated at 0.3 mCE. Regarding the error in modeling the reference pressures, a value of 0.2 mCE is considered realistic. The total uncertainty related to the pressure difference between the measured pressure and the reference pressure is therefore 0.5 mCE.

Various factors including the measuring devices and the state of equilibrium of the network can influence the accuracy of the value of the roughnesses measured in the field, as well as for the estimation of roughness for the other conduits of the network. The following figure 6 presents 3 curves which correspond to the pressure drop caused by a leak in the middle of a pipe without a junction, one with roughness as measured in the field, and two others with limit values taking into account a +/- 10% error related to this parameter ($\mathcal{E}$ + 10%, $\mathcal{E}$ –10%).

**Figure 6 Influence of the error on the roughness of the pipe**

When the roughness is increased, the pressure difference caused by a leak is greater. Conversely, when the pipes are smoother (plastic), the pressure difference decreases. Leak detection by the method studied therefore seems more promising on networks whose hydraulic capacity is weakened by corrosion rates.

**Referee comment**: you have mentioned ". Dynamic simulation is used to describe the operation of the network during a given period, while considering the variation in customers' consumption over time." Physical modeling of your network requires consumption at nodes. It is important to indicate how you estimated this parameter. It should be noted that there is great uncertainty about the nodal distribution of the study area. How did you develop your consumption patterns for dynamic modeling?

Author response: (marked up manuscript lines 135-142)

Author response: (marked up manuscript line 68-70)

**Referee comment**: you have mentioned that you have used the Matlab-software, For EPANET code, please find a better reference

Author response: (marked up manuscript line 193, 440-442,468-470)

For EPANET code, the original work based on:

D.G. Eliades, M. Kyriakou, S. Vrachimis and M.M. Polycarpou, "EPANET-MATLAB Toolkit: An Open-Source Software for Interfacing EPANET with MATLAB", in Proc. 14th International Conference on Computing and Control for the Water Industry (CCWI), The Netherlands, Nov 2016, p.8. (doi:10.5281/zenodo.831493).

Sebbagh, k., Abdelhamid, S., Zabot, M. : Démarche de prélocalisation des pertes physiques sur un réseau de distribution d'eau potable par l'optimisation du modèle hydraulique via un algorithme évolutionnaire, La Houille Blanche., 59-66, doi.org/10.1051/lhb/2016062,2017

685    The number of references were increased to 2.

**Referee comment**: you have mentioned "The model was calibrated without errors" What are the calibration data for your model? Discuss them clearly

690    Author response: (marked up manuscript line 337-343)

**Referee comment**: the calibration results of pressure are presented, you need to complete with the flow calibration

Author response: (marked up manuscript line 341-344)

The calibration is also made against the flow at the three inputs of the study area, the result of the flow simulation (red line)

695    are very close to what is measured green squares as shown in Fig.17.

700

705

710

---

## Author Response (AR2)

[revised manuscript text omitted]

**Point-by-point reply to the comments**

**Topical Editor Decision: Publish subject to minor revisions (review by editor) (18 Jul 2020) by Luuk Rietveld**

Comments to the Author:

Please correct the following:

500    - Avoid inclusion of Figures of others, so only include original ones (thus delete several Figures)

- When (as an exception) a Figure is copied then give reference.

- Refer to Figures as "Figure xxx", NOT "figure xxx" or "Fig. xx"

- increase quality of Figures

- Use always the same name/abbreviation, not "EPANET" and "Epanet" e.g.

505    - write down the algorithm more concised.

**Reply to comments from the Topical Editor:**

The authors would like to thank the topical editor Luuk Rietveld for the time spent helping us to improve this article. The document has been revised, taking into account the comments of the Editor.

**Topical editor comment**: Avoid inclusion of Figures of others, so only include original ones (thus delete several Figures)

510    **Authors comment**: Thank you. The figure 4 (line 116-119) and the figure 11 (264-267) have been deleted.

**Topical editor comment** : When (as an exception) a Figure is copied then give reference

**Authors comment**: This remark is integrated in the revised version of the manuscript. The figure 7 has been updated with reference. (marked up manuscript Lines 183).

515

**Topical editor comment:** Refer to Figures as "Figure xxx", NOT "figure xxx" or "Fig. xx"

**Authors comment**: This remark is integrated in the revised version of the manuscript (marked up manuscript Lines 60,68,73, 157,168,181,234,306,323, 341,342,349,354,362,409, 420)

520    **Topical editor comment:** increase quality of Figures

**Authors comment**: This remark is integrated in the revised version of the manuscript. The quality of figures 2, 4, 5, 8,9, 10,11, 12,13,14, 15 was increased (marked up manuscript Lines 69, 143,160,205,237,311,324,343,350,354,363)

**Topical editor comment**: Use always the same name/abbreviation, not "EPANET" and "Epanet" e.g.

525 **Authors comment**: This remark is integrated in the revised version of the manuscript (marked up manuscript Lines 23,147,189,189;196, 357)

**Topical     editor     comment     :     write     down     the     algorithm     more     concised**
**Authors comment**: This remark is integrated in the revised version of the manuscript. (marked up manuscript lines (272-
530 301)